

# Exploring main soil drivers of vegetation succession in abandoned croplands of Minqin Oasis, China

Li Chang[1,2], Shuhua Yi[3], Yu Qin[2], Yi Sun[3], Huifang Zhang[3], Jing Hu[4], Kaiming Li[1] and Xuemei Yang[5]

[1] School of Environment and Urban Construction, Lanzhou City University, Lanzhou, China
[2] State Key Laboratory of Cryospheric Sciences, Northwest Institute of Eco-Environment and Resources, Chinese Academy of Sciences, Lanzhou, China
[3] Institute of Fragile Eco-environment, School of Geographic Science, Nantong University, Nantong, China
[4] State Key Laboratory Breeding Base of Desertification and Aeolian Sand Disaster Combating, Gansu Desert Control Research Institute, Lanzhou, China
[5] Tourism School, Lanzhou University of Arts and Science, Lanzhou, China

Corresponding author
Li Chang, changli@lzcu.edu.cn

## ABSTRACT

**Background:** The Minqin Oasis, which is located in Wuwei City, Gansu Province, China, faces a very serious land desertification problem, with about 94.5% of its total area desertified. Accordingly, it is crucial to implement ecological restoration policies such as cropland abandonment in this region. In abandoned croplands, abiotic factors such as soil properties may become more important than biotic factors in driving vegetation succession. However, the connections between soil properties and vegetation succession remain unclear. To fill this knowledge gap, this study investigated these connections to explore major factors that affected vegetation succession, which is meaningful to designing management measures to restore these degraded ecosystems.

**Methods:** This study investigated seven 1–29-year-old abandoned croplands using the "space for time" method in Minqin Oasis. Vegetation succession was classified into different stages using a canonical correlation analysis (CCA) and two-way indicator species analysis (Twinspan). The link between soil properties and vegetation succession was analyzed using CCA. The primary factors shaping community patterns of vegetation succession were chosen by the "Forward selection" in CCA. The responses of dominant species to soil properties were analyzed using generalized additive models (GAMs).

**Results:** Dominant species turnover occurred obviously after cropland abandonment. Vegetation succession can be classified into three stages (*i.e.*, early, intermediate, and late successional stages) with markedly different community composition and diversity. The main drivers of vegetation succession among soil properties were soil salinity and saturated soil water content and they had led to different responses of the dominant species in early and late successional stages. During the development of vegetation succession, community composition became simpler, and species diversity decreased significantly, which was a type of regressive succession. Therefore, measures should be adopted to manage these degraded, abandoned croplands.

# INTRODUCTION

Vegetation succession, a topic of interest in the context of ecological restoration (*Janečková et al., 2021*; *Coradini, Krejčová & Frouz, 2022*), is often studied using the space-for-time substitution method (*Zhang et al., 2021*; *Hao & Dong, 2023*). The principles of spontaneous vegetation succession can be used as a scientific basis for restoration ecology, including knowledge of plant community composition, species diversity, species turnover, and driving factors. These learnings can guide land managers in restoring damaged ecosystems (*Arif et al., 2022*; *Ding et al., 2022*; *Gul et al., 2022*; *Hu et al., 2022*). Specifically, abandoned croplands can provide a classic example for vegetation succession studies.

Numerous studies have been implemented to investigate vegetation succession in abandoned fields, which is a type of secondary succession that follows anthropogenic disturbance (*Albert et al., 2014*; *Prach, Jírová & Doležal, 2014*; *Martínez-Ramos et al., 2021*; *Zivec et al., 2021*; *Coradini, Krejčová & Frouz, 2022*; *Moyo & Ravhuhali, 2022*; *Yan et al., 2023*). In studies of secondary succession, most researchers highlight the importance of biotic factors such as seed bank, seed dispersal, competitive ability, arrival order of plant species during succession, plant-animal dynamics, and biofeedback within soil microbial communities (*Török et al., 2018*; *Horáčková, Řehounková & Prach, 2019*; *Prach & Walker, 2019*; *Mudrák et al., 2021*; *Zhang et al., 2023*; *Wang et al., 2023*). However, other researchers believe that abiotic factors during the secondary vegetation succession process should receive more attention. For example, soil nitrogen may have more significant effects on vegetation succession than soil phosphorus at the early and middle successional stages, while the reverse is found at the late successional stages (*Batterman et al., 2013*). Soil moisture (*Prach, Jírová & Doležal, 2014*; *Huang et al., 2017*) is considered an important driver of vegetation succession. In addition, high saline-alkaline stress can limit plant community succession (*Bai et al., 2015*). However, it is still unclear about abiotic drivers of vegetation succession in abandoned croplands of arid regions.

Arid areas experience extreme environmental conditions. For example, the Minqin Oasis is one of the four major source regions of sand storms in China due to severe land desertification, which covers 94.5% of its total area (*Zhao, Yang & Zhu, 2018*). Specifically, soil salinization is the primary form of soil desertification in the Minqin Oasis, with over 80% of the total land in the region considered salinized land (*Yang et al., 2020*). Furthermore, soil desertification has aggravated due to over-farming and the excessive exploitation of underground water (*Niu et al., 2016*). Accordingly, since 2006, Minqin County adopted the "Grain-for-Green" project, which involved restoring cropland to perennial vegetation including grassland, shrubland, and forest (*Wang et al., 2020*).

Abandoned croplands of soil salinization are only scattered in the Hu area, which is one of the three irrigated regions in the Minqin Oasis (*Wang et al., 2022a*). As it is located at the tail-end of the Shiyang River Basin, the Hu area plays a key ecological role in preventing the consolidation of the Badain Jaran Desert and the Tengger Desert. The Hu area is extremely fragile and exhibited a significant increase in salinization from 2015 to

2018 (*Yang et al., 2020*). Meanwhile, the abandoned croplands in the Hu area represent both a chronological sequence and a soil salinity gradient. Therefore, the secondary succession of the abandoned croplands in the Hu area may be viewed as primary succession. Accordingly, abiotic factors, especially soil properties such as soil salinity, may have an even greater effect than biotic factors on vegetation succession. Nonetheless, these connections were merely reported with correlation analysis in previous studies, which is difficult to find the main drivers (*He et al., 2021*; *Wang et al., 2022b*).

To fill the above-mentioned knowledge gap, this study focused on the effects of soil properties on vegetation succession using multivariate analysis. This study aims to: (1) characterize shifts in dominant species after cropland abandonment; (2) explore primary factors that can drive vegetation succession among soil properties; and (3) analyze responses of dominant species in the early and late successional stages to the primary factors. The findings of this study may play an important role in abandoned cropland management and desertification prevention in arid and semiarid areas.

## MATERIALS AND METHODS

### Study site

The Minqin Oasis (38°05′–39°27′N; 101°48′–104°13′E) is surrounded by the Tengger and Badain Jaran deserts and is located along the lower reaches of the Shiyang River in Wuwei City, Gansu Province, China (Fig. 1). This oasis is mainly characterized by low mountains along with the Gobi and other deserts, with elevations in the Minqin Oasis ranging from 1,249 to 2,000 m. This area has a typical temperate desert climate with an average annual temperature of 7.6 °C and an average annual precipitation of 113.2 mm. Gray-brown desert soil is the main zonal soil (*Wang et al., 2021*). In addition, meadows, saline-alkali, and aeolian sandy soils occur in some areas. The natural vegetation types mainly include temperate desert vegetation, temperate desert steppe vegetation, deciduous broad-leaved brush, and desert meadow (*Liu et al., 2006*).

The Minqin Oasis runs nearly 140 km north to south and 40 km east to west and was developed under the long-term effects of the ancient Shiyang and Jinchuan rivers (*Feng, 1963*). The oasis has been divided into three irrigated areas, specifically the Ba, Quanshan, and Hu areas, based on geographical location, the farming methods used in the area, and the design of the aqueducts and canals (*Xu & Cheng, 2005*; *Zhang et al., 2015*).

### Field sampling design

Seven 1–29-year-old abandoned croplands in the Hu area were selected for sampling in late August 2017 (Table 1). These sampling fields were discontinuously (separated by at least 130 m) located in a flat alluvial fan (1,249 m a.s.l.) with areas ranging from 60 m × 60 m to 100 m × 100 m (measured based on the boundaries of the original croplands). In each sampling field, three 10 m × 10 m plots were selected to sample shrubs on the diagonal of each field (Fig. 2). In each 10 m × 10 m plot, three 1 m × 1 m subplots were used to sample herbaceous vegetation (Fig. 2). The sampling field (f1) was 1 year old and composed of three parts because larger young abandoned cropland was difficult to find (Table 1). In each part of the

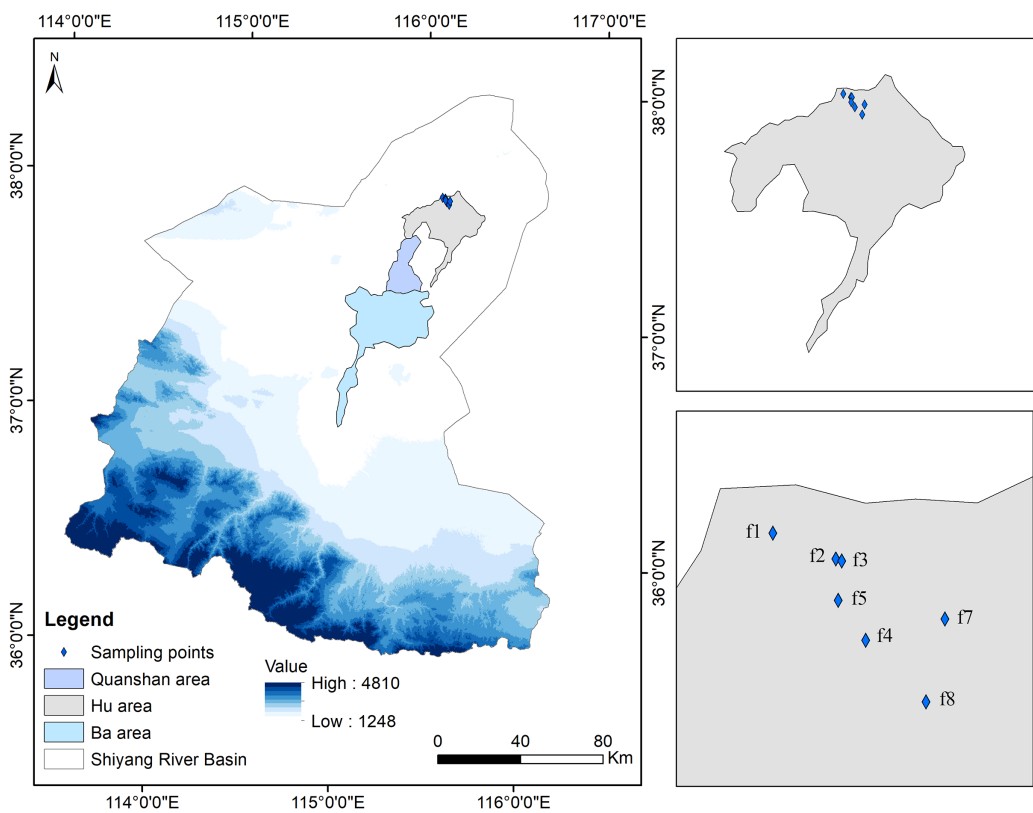

**Figure 1 Map of sampling fields in abandoned croplands of the Minqin Oasis.** Seven abandoned croplands are labeled f1 to f7 in the Minqin Oasis. Years since land abandonment appear in parentheses following labels, f1 (1 year old), f2 (6 year old), f3 (7 year old), f4 (8 year old), f5 (9 year old), f6 (20 year old) and, f7 (29 year old).

**Table 1 Basic information on abandoned croplands.**

| Code | Longitude and latitude | | Years since land abandonment (year old) |
|------|------------------------|--|------------------------------------------|
| f1 | 103.59°E | 39.06°N | 1 |
| f2 | 103.60°E | 39.06°N | 6 |
| f3 | 103.60°E | 39.06°N | 7 |
| f4 | 103.60°E | 39.04°N | 8 |
| f5 | 103.60°E | 39.05°N | 9 |
| f6 | 103.62°E | 39.03°N | 20 |
| f7 | 103.63°E | 39.04°N | 29 |

field, one 10 m × 10 m plot was established to sample shrubs and herbaceous vegetation because a 1 m × 1 m subplot was too small to sample larger herbaceous vegetation.

## Vegetation data

Two kinds of data were used in this study: species composition data (matrix **C,** 21 plots × 33 species) and plant association data. These raw data were collected from twenty-one
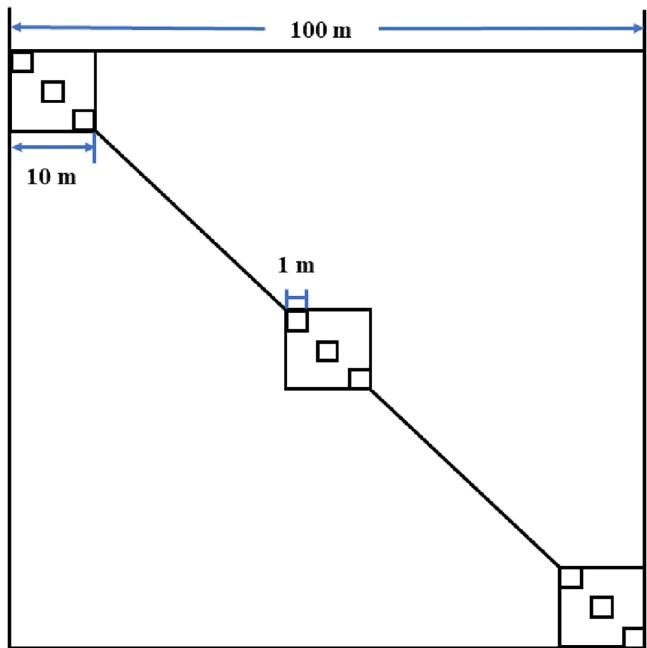

**Figure 2** **Field sampling design in abandoned croplands of the Minqin Oasis.**

100-m² plots and fifty-four 1-m² subplots, where crown breadth, height, and abundance were recorded.

Species composition data were transformed into presence/absence data to lower the weights of rare species (*Zelený & Chytrý, 2019*). Every set of three 1-m² herbaceous subplots was then transformed into one 100-m² shrub plot, and the presence/absence data were used to implement multivariate analysis. Plant association data were used to select several dominant species to represent large numbers of individuals and simplify the plant-soil relationship. These data were calculated using the following steps. First, two-way indicator species analysis (Twinspan) was used to classify twenty-one 100-m² plots into seven groups that represented seven plant associations. Then, the average crown breadths, heights, and abundances of each herbaceous species were calculated among three nested 1-m² subplots within a 100-m² plot. The herbaceous abundance was expanded 100 times because of its dependence on plot area; all other data remained the same. The herbaceous data were then transformed from 1-m² subplots to 100-m² plots. Finally, the importance values of herbaceous and shrub species were evaluated together for each 100-m² plot, and then individually for each plant association.

## Soil data

The soil data were arranged as a matrix of 21 plots × nine factors (matrix **S**), containing the saturated soil water content (SSW, %), soil field capacity (SFC, %), soil organic matter (SOM, %), total nitrogen (TN, %), available phosphorus (AP, mg/100 g soil), pH, total salinity (TS, %), electrical conductivity (EC, μm/cm), and years since land abandonment (year, year old). These data were collected and determined using the following methods.

In each 100-m$^2$ plot (Fig. 2), three replicates were sampled to collect two types of surface soil samples (0–10 cm). One type of sample was collected with circular knives near a 1-m$^2$ subplot, and the other type of soil sample was a mixed sample collected from five locations within a 1-m$^2$ subplot. The former sample was used to determine SSW and SFC with the cutting ring method (*Xu, 2018*), and the latter sample was used to analyze the soil chemical properties after the samples were air-dried and sieved through a 2-mm screen. The following soil properties were determined using conventional methods (*Bao, 2000*): SOM (potassium dichromate oxidation method), TN (Kjeldahl method), AP (molybdenum antimony colorimetric method), pH, TS (sum of ions), and EC (electrode method). The age of the seven abandoned croplands (Year) was provided by the Minqin County Forestry Bureau and local residents.

## Data analysis

This study categorized different types of successional stages and revealed the links between vegetation succession and soil properties using multivariate analyses in the Canoco and Pcord computer programs (*Mccune & Grace, 2002*; *del Moral, 2009*; *del Moral, Sandler & Muerdter, 2009*; *Peck, 2010*). Twenty-one plots were grouped into seven plant associations based on species composition data (**C**), using Twinspan to facilitate analysis and interpretation (*Zanini et al., 2014*). Then, the seven plant associations were classified into three successional stages based on the same dataset (**C**), considering the diagram of seven plant associations in the canonical correlation analysis (CCA) ordination space. CCA was further performed with species composition data (**C**) and soil data (**S**) to explore the general trends of vegetation succession and soil properties. Forward selection was used in CCA to choose the primary factors shaping the community patterns in vegetation succession with a false discovery rate (FDR) to improve the accuracy of the results (*Verhoeven, Simonsen & McIntyre, 2005*). The significance of the first axis ($P = 0.0062$) and all axes ($P = 0.0002$) of CCA were tested with 9,999 permutations.

Species response curves were used to summarize the responses of seven dominant species to the specific soil factors, based on generalized additive models (GAMs) using species composition data (**C**) and soil data (**S**). In the GAM options, a value of four was specified in the term smoothness field, Gaussian or Poisson distribution was selected, and the lowest Akaike Information Criterion value (AIC) was used to select the most adequate model (*Rodríguez, Otto & Fernández-Palacios, 2022*).

Plant diversity was analyzed using species richness (N), Shannon–Weiner Index (H'), Pielou's J index (E), Simpson Index (D), and Importance Value ($p_i$), which were calculated using the following equations: (*Karen et al., 2004*; *Bello, Leps & Sebastià, 2006*; *Bonham, 2013*; *Spellerberg & Fedor, 2003*; *Wesuls et al., 2013*; *Sun, Yi & Hou, 2018*).

$$N = species\ number\ in\ each\ plot \tag{1}$$

$$H' = -\sum p_i \ln p_i \tag{2}$$

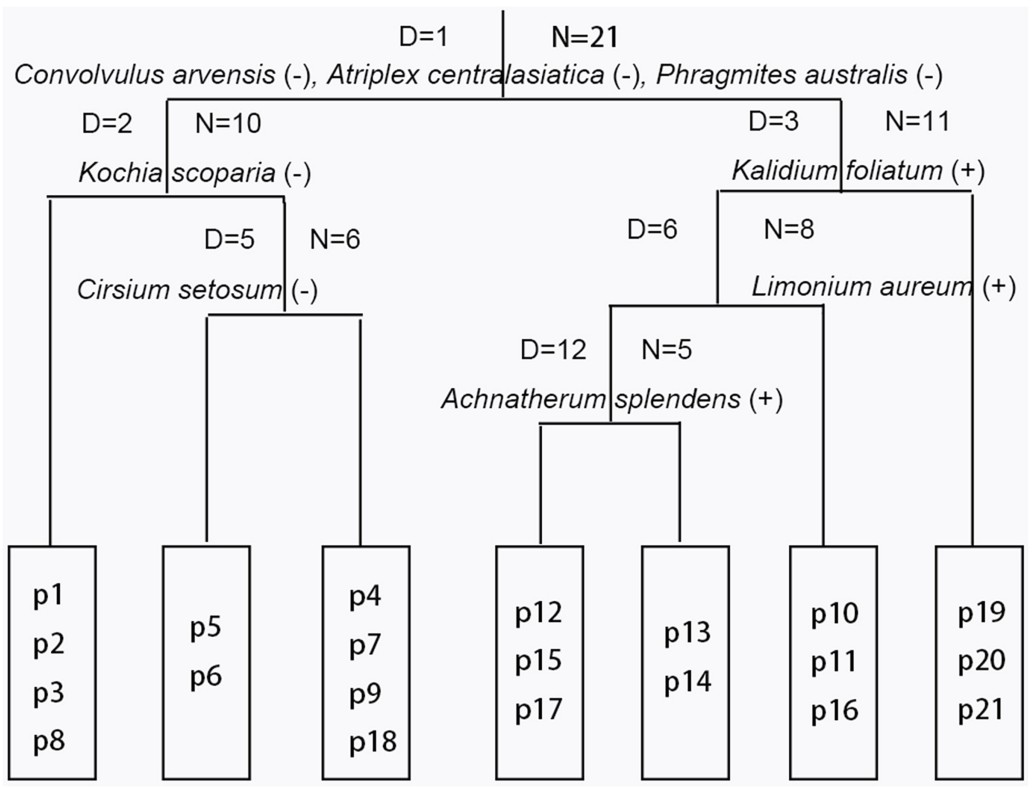

**Figure 3 Results of two-way indicator species analysis (Twinspan).** $D = n$ ($n$ = 1, 2, 3, …): the $n^{th}$ division; $N = n$ ($n$ = 1, 2, 3, …): there are n plots before division; the species name: the indicator species in this division; (+) or (−): the indicator species belongs to the positive group or the negative group, respectively.

$$E = H' / \ln(N) \tag{3}$$

$$D = 1 - \sum p_i^2 \tag{4}$$

$$p_i = \frac{C_r + H_r + D_r}{3} \tag{5}$$

where $C_r$ represents the relative coverage, $H_r$ represents the relative height, and $D_r$ stands for the relative density. The differences in species diversity indices among successional stages were identified using Kruskal–Wallis tests and the boxplots were plotted using the "boxplerk" function in R (*R Core Team, 2020*; *Borcard, Gillet & Legendre, 2020*).

## RESULTS

### The division of vegetation succession stages

The results from Twinspan and the CCA analysis indicated that vegetation succession in the Minqin Oasis was composed of early, intermediate, and late successional stages (Figs. 3, 4). The early successional stage (I) contained an *Atriplex centralasiatica* association (assoc. 1), and the later stage (III) contained a *Kalidium foliatum* association (assoc. 7; Figs. 3, 4). The intermediate stage (II) contained the remaining associations (Figs. 3, 4). Vegetation

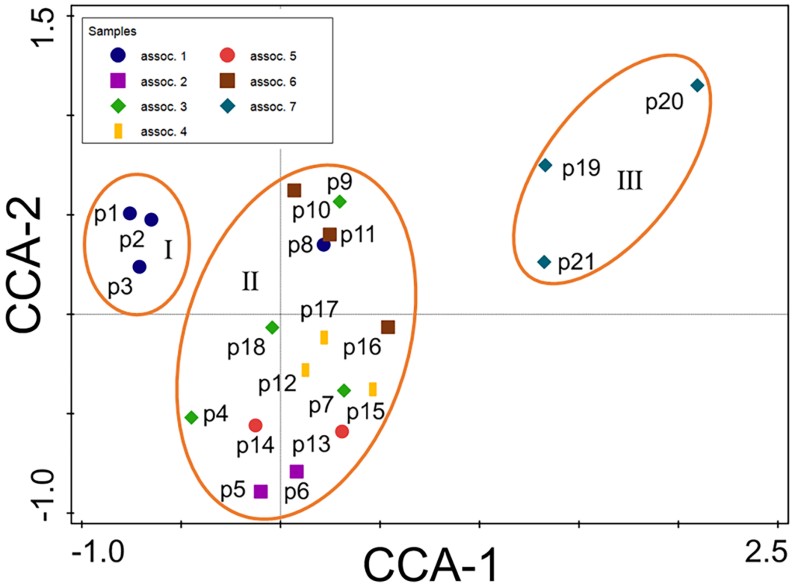

**Figure 4 Canonical correlation analysis (CCA) ordination diagram of 21 plots.** Assoc. 1–7: seven plant associations divided using Twinspan; assoc. 1: *Atriplex centralasiatica*; assoc. 2: *Peganum nigellastrum*; assoc. 3: *Peganum nigellastrum—Tamarix chinensis*; assoc. 4: *Peganum nigellastrum—Lycium chinense*; assoc. 5: *Nitraria tangutorum—Halogeton glomeratus*; assoc. 6: *Halogeton glomeratus—Tamarix chinensis*; assoc. 7: *Kalidium foliatum*.

succession progressed from the *A. centralasiatica* association to the *Peganum nigellastrum* association, *Nitraria tangutorum—Halogeton glomeratus* association, *P. nigellastrum—Tamarix chinensis* association, *H. glomeratus—T. chinensis* association, *P. nigellastrum—Lycium chinense* association, and finally to the *K. foliatum* association in the abandoned cropland of the Minqin Oasis. Specifically, succession evolved from an annual herb association to either a perennial herb or annual (perennial) herb—shrub association, and finally to a subshrub association.

## Community composition and diversity

The community composition was markedly different in the early, intermediate, and late successional stages. The early successional stage (aged 1–7 year old) contained only one plant association (assoc. 1) but had the largest number of species per plant association among the three successional stages (24 species per plant association; Table 2). The importance value (IV) of herbs was larger than that of shrubs. As a result, the annual herbaceous plant *A. centralasiatica* became the dominant species and the perennial herb *P. nigellastrum* became the most common plant species in this stage (Table 2). The late successional stage (aged 29 year old) also contained only one plant association (assoc. 7), this association only included four species. The shrub layer was more dominant than the herb layer, and one subshrub, *K. foliatum*, was the monodominant species (Table 2). The intermediate successional stage (aged 6–20 year old) contained five plant associations (assoc. 2–6), which were composed of mixtures of dominant herbaceous species and shrub species, except for assoc. 2 (Table 2). This finding showed that the dominance of the shrub layer increased and that of the herb layer gradually decreased during the intermediate successional stage (Table 2).

**Table 2 Importance values of plant species in seven associations.**

| Life form | Species | Importance value(%) | | | | | | |
|---|---|---|---|---|---|---|---|---|
| | | assoc. 1 | assoc. 2 | assoc. 3 | assoc. 4 | assoc. 5 | assoc. 6 | assoc. 7 |
| Herb | *Halogeton glomeratus* | 9.00 | 0.60 | 4.40 | 23.95 | **26.19**[*] | **37.96** | 0.59 |
| | *Kochia scoparia* | 4.05 | | | | | | |
| | *Convolvulus arvensis* | 2.29 | 2.32 | 0.14 | | | | |
| | *Atriplex centralasiatica* | **38.25** | 3.96 | 9.30 | | | | |
| | *Chloris virgata* | 1.06 | | | | | | |
| | *Suaeda glauca* | 5.00 | | | | | | |
| | *Chenopodium album* | 3.79 | | | | | | |
| | *Phragmites australis* | 2.30 | | 0.74 | | | | |
| | *Mulgedium tataricum* | 0.65 | 1.19 | | | | | |
| | *Cynanchum sibiricum* | 0.86 | | | | | | |
| | *Euphorbia humifusa* | 0.07 | | | | | 2.05 | |
| | *Peganum harmala* | 0.78 | 0.72 | 0.95 | | 1.32 | 2.05 | |
| | *Glycyrrhiza uralensis* | 0.84 | | | | | | |
| | *Achnatherum splendens* | 2.80 | | | | 13.87 | | |
| | *Suaeda prostrata* | 0.29 | | | | | | |
| | *Setaria viridis* | 0.48 | | | | | | |
| | *Cardaria chalepensis* | 0.15 | | | 8.96 | | | |
| | *Peganum nigellastrum* | 20.51 | **53.67** | **38.02** | **26.62** | 15.67 | 2.77 | 10.95 |
| | *Leymus secalinus* | | 1.82 | | | | | |
| | *Cirsium setosum* | | 2.22 | | | | | |
| | *Bassia dasyphylla* | | | | 7.78 | | 0.40 | |
| | *Limonium aureum* | | | | | | 21.48 | |
| | *Acroptilon repens* | | | | 4.26 | 2.83 | | |
| | *Cynanchum chinense* | | | 1.00 | | | | |
| | *Echinochloa crusgalli* | 0.60 | | | | | | |
| Shrub | *Lycium chinense* | 0.82 | 5.40 | 4.79 | **20.75** | 5.61 | | |
| | *Tamarix chinensis* | 1.50 | | **22.40** | | | **24.27** | |
| | *Lycium ruthenicum* | 1.74 | 3.99 | 4.86 | 7.69 | 4.50 | 9.01 | 17.81 |
| | *Nitraria tangutorum* | | 9.29 | 0.78 | | **29.99** | | |
| | *Nitraria sibirica* | 1.35 | 2.51 | 2.22 | | | | |
| | *Reaumuria songarica* | | 2.13 | 1.50 | | | | |
| | *Tamarix hispida* | | 10.13 | | | | | |
| | *Kalidium foliatum* | 0.83 | | 8.91 | | | | **70.66** |

**Note:**
[*] The highest importance value for each assoc. is indicated in bold.

The results of the Kruskal–Wallis test indicated that plant diversity among successional stages showed significant differences except for Pielou's J index (E; Fig. 5). The largest values of the Shannon–Weiner Index (H'), Simpson Index (D), and species richness ($\alpha$) were 2.20, 0.85, and 13.75, respectively, which all occurred during the early successional stage. H', D, and $\alpha$ then declined over time from the early successional stage to the late successional stage (Fig. 5).

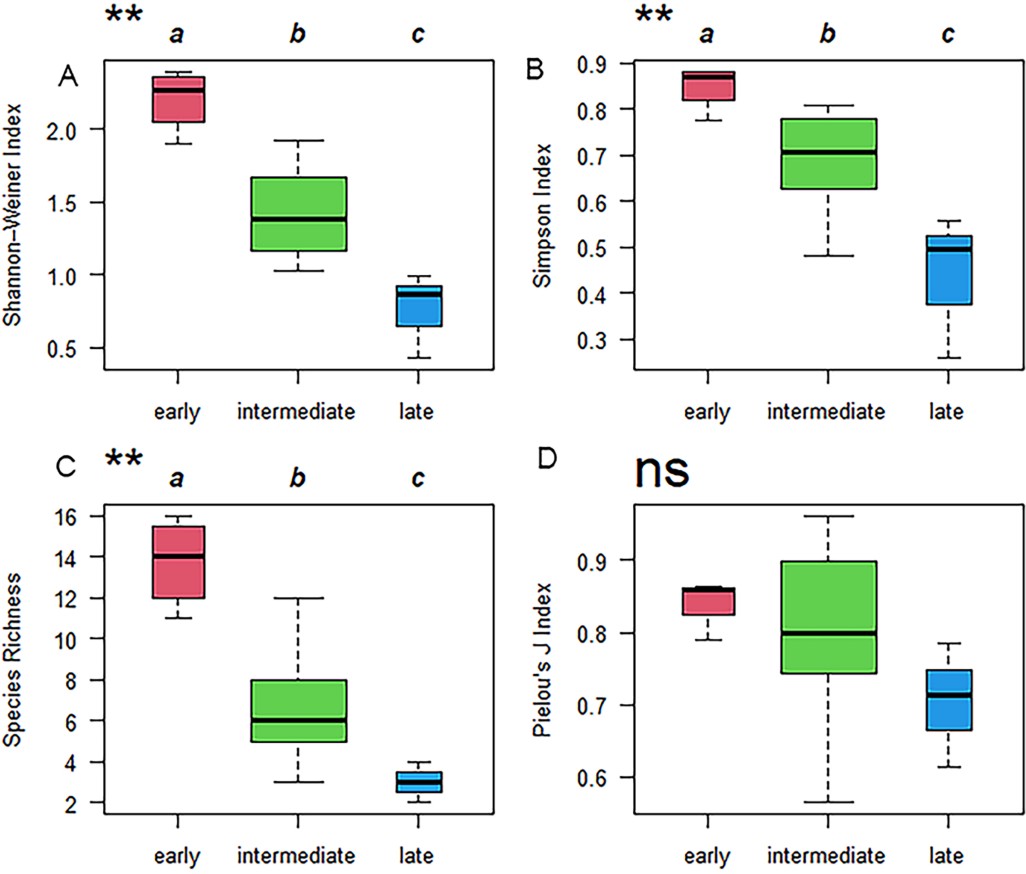

**Figure 5 Boxplot of plant diversity, analyzed using the Kruskal–Wallis test during the early, intermediate, and late successional stages.** (A) Shannon–Weiner Index, (B) Simpson Index, (C) species richness, and (D) Pielou's J Index. The same lowercase letters (a, b, and c) outside the box indicate that there was no significant difference between the two successional stages, while different lowercase letters indicate a significant difference. ** represents significance at $P < 0.01$. "ns" means no significance.

## The link between vegetation succession and soil properties

The 21 plots were regularly distributed and divided into three successional stages along the first CCA axis (Fig. 4). The first CCA axis ($P < 0.05$) represents the direction of vegetation succession from left to right and indicates a gradient of increasing successional time. This pattern of vegetation succession can be interpreted with only two soil properties using forward selection (Figs. 4, 6; Table 3). The most important factor was TS, which explained 13.3% of the total variation, followed by SSW (9.0%) in the CCA ordination (Fig. 6; Table 3).

Figure 7 further confirms the causal link between vegetation succession and soil properties. The figure suggests that the responses of the dominant plant species during the early and late successional stages—*A. centralasiatica* and *K. foliatum*, respectively—to the main soil properties (TS and SSW), were markedly different (Figs. 7F, 7I). The early dominant species had a bell-shaped response to increases in SSW ($R^2 = 48.8$, AIC = 31.20),

Chang et al. (2024), *PeerJ*, DOI 10.7717/peerj.17627      10/22

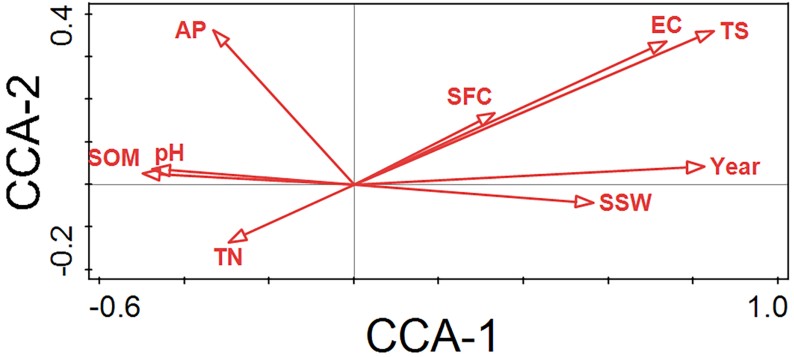

**Figure 6 CCA ordination diagram of environmental variables.** SOM, soil organic matter; AP, available phosphorus; EC, electrical conductivity; TN, total nitrogen; SSW, saturated soil water content; SFC, soil field capacity; TS, total salinity; Year, years since land abandonment.

**Table 3 Simple and conditional effects of explanatory variables.**

| Name | Explains % | Pseudo-F | P | P (adj) |
|---|---|---|---|---|
| Simple term effects: | | | | |
| TS | 13.3 | 2.9 | 0.0001 | 0.0009 |
| Year | 11.9 | 2.6 | 0.0003 | 0.00135 |
| EC | 11.0 | 2.4 | 0.0018 | 0.0054 |
| SSW | 9.0 | 1.9 | 0.0085 | 0.01913 |
| AP | 7.6 | 1.6 | 0.0574 | 0.08859 |
| SFC | 7.4 | 1.5 | 0.0689 | 0.08859 |
| SOM | 7.3 | 1.5 | 0.0653 | 0.08859 |
| pH | 5.5 | 1.1 | 0.3143 | 0.35359 |
| TN | 5.1 | 1.0 | 0.4172 | 0.4172 |
| Conditional term effects: | | | | |
| TS | 13.3 | 2.9 | 0.0001 | 0.0009 |
| SSW | 9.0 | 2.2 | 0.0078 | 0.0351 |
| AP | 7.7 | 2.0 | 0.0188 | 0.0564 |
| SFC | 7.4 | 1.7 | 0.0372 | 0.0837 |
| EC | 5.0 | 1.3 | 0.1814 | 0.31243 |
| SOM | 4.8 | 1.3 | 0.2095 | 0.31243 |
| Year | 4.5 | 1.2 | 0.243 | 0.31243 |
| pH | 2.5 | 0.7 | 0.7781 | 0.87536 |
| TN | 2.0 | 0.5 | 0.9053 | 0.9053 |

**Note:**
$P$, level of significance of explanatory variables; $P$ (adj), adjusted level of significance of explanatory variables by using the false discovery rate approach; SOM, soil organic matter; AP, available phosphorus; EC, electrical conductivity; TN, total nitrogen; SSW, saturated soil water content; SFC, soil field capacity; TS, total salinity; Year, years since land abandonment.

and it decreased with an increase in TS ($R^2$ = 24.30, AIC = 32.35) (Figs. 7F, 7I). In contrast, the late dominant species rapidly increased as SSW ($R^2$ = 18.4, AIC = 26.38) and TS ($R^2$ = 37.70, AIC = 23.61) increased (Figs. 7F, 7I). Therefore, the late dominant species replaced the early dominant species in the late successional stage.

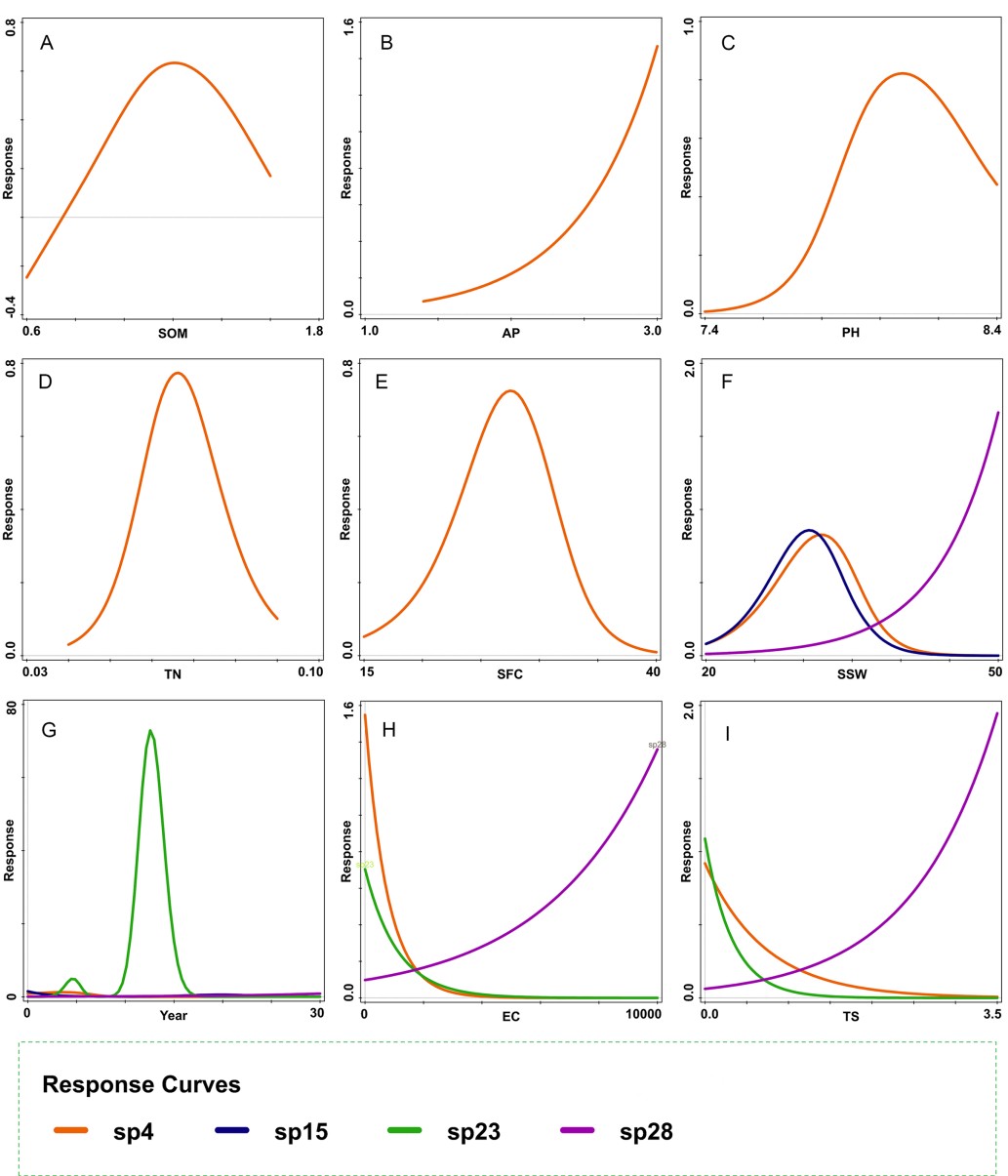

**Figure 7 Response curves of dominant species fitted by generalized additive models.** (A) SOM, (B) AP, (C) PH, (D) TN, (E) SFC, (F) SSW, (G) Year, (H) EC, and (I) TS. sp4: *Atriplex centralasiatica;* sp15: *Tamarix chinensis;* sp23: *Nitraria tangutorum;* sp28: *Kalidium foliatum*. SOM, soil organic matter; AP, available phosphorus; TN, total nitrogen; SFC, soil field capacity; SSW, saturated soil water content; Year, years since land abandonment; EC, electrical conductivity; TS, total salinity.

## DISCUSSION

### Process of vegetation succession

The results of this study agree with previous findings that the successional direction moves from annual herb to subshrubs in the abandoned croplands of the Minqin Oasis (*He et al., 2010*; *Li et al., 2010, 2011*; *Wang et al., 2013*; *Yan et al., 2014*; *Wang, 2016*; *Ma et al., 2018*).

However, some differences exist in the succession pathways. Some studies have revealed that vegetation succession begins with the colonization of newly-abandoned croplands by farmland weeds (*He et al., 2010*; *Li et al., 2010*; *Li et al., 2011*; *Wang et al., 2013*; *Yan et al., 2014*; *Wang, 2016*) such as *Chenopodium album*, *Convolvulus arvensis*, *A. centralasiatica*, and *Kochia scoparia*, and then ends with the colonization of halophytic plants, which are dominant species of solonchak desert communities in the Minqin Oasis (*Liu, 2008*), such as *K. foliatum* (*He et al., 2010*; *Wang et al., 2013*; *Wang, 2016*) or *Kalidium cuspidatum* (*Li et al., 2010*; *Yan et al., 2014*). Other studies have indicated that a pioneer species, *H. glomeratus*, initially invades abandoned cropland after 1 year (*Ma et al., 2018*). *H. glomeratus* is not a typical farmland weed species, but is one of the dominant species of plant associations of annual desert plants (*Peng et al., 2004*). A subshrub, *Corethrodendron scoparium*, which is not a halophytic plant, gradually replaces the annual herbs in the late successional stages (*Ma et al., 2018*). These two distinct successional pathways may arise in response to different soil properties.

The results of this study also showed that plant diversity significantly decreases during the process of vegetation succession in the Minqin Oasis, which is consistent with the results of *Baba, Tanaka & Kusumoto (2019)*. However, *Heydari et al. (2020)* pointed out that the species biodiversity of all functional groups increases significantly over time as recently abandoned croplands develop into forests. It is evident that soil properties may induce this type of difference. In our study region, gray-brown desert soil is the main zonal soil. Irrational irrigation methods and a lowering of the ground-water table may lead to soil salinization, resulting in a rapid decrease in species diversity. *Heydari et al. (2020)* demonstrated that soil fertility improvement after long-term land abandonment increases species diversity. In addition, research has shown that with the progress of succession, biodiversity may exhibit: an unimodal trend, different trends for different functional groups, or no directional trend (*Sun et al., 2017*; *Jin et al., 2021*; *Chen et al., 2021*; *Sharma, Kumar & Ovung, 2022*).

## Drivers of vegetation succession

The results of this study suggest that TS is the main factor driving the development of vegetation succession in the abandoned croplands of the Minqin Oasis, followed by SSW. A high TS value means many soluble salts have accumulated near the soil surface, leading to the development of secondary salinization; in agriculture, this is often caused by irrational irrigation methods (*Rengasamy, 2006*). As a type of stress affecting plant growth, soil salinity may influence nutrient absorption, N-fixing symbioses, photosynthesis efficiency, osmotic potential, and ion toxicity (*Läuchli & Lüttge, 2002*; *Parida & Das, 2005*; *Grieve, Grattan & Maas, 2011*). Generally, the level of plant stress increases with an increase in soil salt. High salt stress filters out salt-intolerant plant species, helping shape community patterns in vegetation succession (*Bui, 2013*).

In addition to soil salt content, *Liu et al. (2017)* suggested that the water table may also play a vital role in vegetation succession in Minqin. One main difference between this previous study and the current study is in the methods of human disturbance. The study by *Liu et al. (2017)* was conducted in the area of Qingtu Lake, which dried up in 1959

(*Chunyu et al., 2019*). The government refilled this lake starting in 2010, and its area had expanded to 25.16 km$^2$ by 2016 (*Liu et al., 2017*). Large amounts of added water dramatically affected natural vegetation succession around Qingtu Lake. The present study was conducted in abandoned croplands, which were cultivated by farmers in the past, and the turnover of dominant species was significant as the TS increased between the early and late successional stages. A recent study found that the variation of soil water content in the 0–40 cm soil layer was not significant between the early (aged 1 year old) and late (aged 30 year old) successional stages (*He et al., 2021*), indicating that the influence of soil salt content may be greater than that of water.

SSW was the second main factor affecting succession. The SSW is the theoretical maximum value of soil water content; the real soil water content value is often lower than the SSW in arid areas. Although the SSW cannot reflect the actual soil water content value, it can generally measure the looseness of the soil layer. This means that the shallow soil layer was looser in the late successional stage than in the early successional stage in the abandoned croplands of the Minqin Oasis. This may also be an indirect cause of species turnover. The root depth of herbaceous species is always shallower than that of shrub or subshrub species. In this study, the most dominant species during the early and intermediate successional stages were herbaceous species. However, in the late successional stage, the subshrub *K. foliatum* replaced herbaceous species. The number of herbaceous species declined from 25 to two, and their IV value was very low (Table 2). This may be because the herbaceous species cannot firmly fix their roots in the shallow soil layer as the looseness of the soil layer increases.

Although SOM is a very important index used to indicate soil quality and nutrient conditions (*Gu et al., 2019*), SOM did not significantly affect vegetation succession in this study. This may be because the fluctuating trend of the SOM could not explain the species turnover from the early to late successional stages. The results showed that the SOM content decreased with time since agricultural abandonment. However, extensive studies have shown that the SOM content in abandoned cropland exhibits the opposite trend (*Li et al., 2020*; *Valverde-Asenjo et al., 2020*; *Rong et al., 2021*; *Xu et al., 2021*; *Zheng et al., 2023*). This decrease in SOM may be because the loss of plant species diversity (Fig. 5) decreases the quantity of litter and roots, which are the main resources of SOM (*Wang et al., 2018*; *Xu et al., 2021*).

### Mode of vegetation succession

The results of this study indicate that soil properties may dominate the direction of vegetation succession in the abandoned croplands of the Minqin Oasis. Vegetation succession in the Minqin Oasis is an allogenic factor-driven type of succession in which species turnover can be achieved through an autogenic process of plant-to-plant interaction (*Hobbie, 1992*), meaning species replacement can be predicted based on soil properties.

The conclusion of this study is supported by two specific findings. Firstly, this study demonstrated that vegetation succession could be predicted based on soil properties because the first axis ($P = 0.012$) and all axes ($P = 0.0007$) of CCA successfully passed the

Monte Carlo test (Figs. 4, 6). The first two axes of CCA explained 27.40% of the total variation, indicating that the link between vegetation succession and soil properties in CCA ordination is reliable (Figs. 4, 6). Forward selection further illustrated that TS and SSW significantly affect vegetation succession (Table 3). Similarly, the species response curves of dominant species in plant associations demonstrated a causal link between dominant species and soil properties (Fig. 7). The dominant species thus changed from the early successional stage to the late successional stage as main soil properties significantly changed.

Second, the trends of species turnover and soil properties reconfirmed the first finding. In the early successional stage, species randomly colonized newly-abandoned cropland. In this stage, the habitat conditions were still relatively good: the soil was rich in nutrients and the salinity was low. Once farm crops were abandoned, many weeds invaded, such as *K. scoparia*, *C. arvensis*, *A. centralasiatica*, *Chloris virgata*, *C. album*, *Phragmites australis*, *Setaria viridis*, and *Cirsium setosum*. Environmental filtering does not play an important role in initial colonization (*del Moral & Eckert, 2005*; *del Moral, Sandler & Muerdter, 2009*). In the late successional stage, the number of species had fallen sharply from 24 to four (Table 2), with more than half of the early herbaceous plants disappearing in the old field. Further, species composition became very simple with only a monodominant association of *K. foliatum*, which is usually the dominant species of solonchak desert communities (*Liu, 2008*). TS and SSW also increased from the early to the late successional stage (Figs. 4, 6 and 7I), which led to the establishment of *K. foliatum* associations because this species can adapt to a relatively high level of soil salt and poor soil quality.

## Limitations and prospects

This study has found the main soil drivers of vegetation succession in the Minqin Oasis, which is a representative arid region. However, there are still some limitations that need further investigation in the future. First, the universality of the findings in this study needs further verification since we only investigated in a small region. Second, only soil properties were analyzed in this study, and other abiotic factors can be explored in the future. In addition, our analysis mainly used traditional statistical methods, which may ignore some potential correlations. More advanced machine-learning methods can be used in future work.

## CONCLUSIONS

Secondary vegetation succession in the Minqin Oasis is a type of allogenic vegetation succession induced by soil properties. The main drivers of vegetation succession in the abandoned croplands in the oasis were total salinity (TS) and saturated soil water content (SSW). During the process of secondary vegetation succession, the species composition became simpler as the plant community changed to adapt to the higher level of soil salt and poor soil quality. This passive restoration without human intervention in the Minqin Oasis is the result of regressive succession. Management measures using the link between vegetation succession and soil properties should be implemented to restore degraded abandoned croplands.

## ACKNOWLEDGEMENTS

The authors would like to thank Delu Li and Fanglan He from the Gansu Desert Control Research Institute for assistance with plant species identification, but also to thank Tao Zhang, Marel Stock and LetPub for linguistic assistance during the preparation of this manuscript.

### Funding

This research was funded by the National Natural Science Foundation of China (grant number 41801102, 41661014), and the Doctoral Research Fund of Lanzhou City University (LZCU-BS2020-05). The funders had no role in study design, data collection and analysis, decision to publish, or preparation of the manuscript.

### Grant Disclosures

The following grant information was disclosed by the authors:
National Natural Science Foundation of China: 41801102, 41661014.
Doctoral Research Fund of Lanzhou City University: LZCU-BS2020-05.

### Competing Interests

The authors declare that they have no competing interests.

### Author Contributions

- Li Chang conceived and designed the experiments, authored or reviewed drafts of the article, and approved the final draft.
- Shuhua Yi conceived and designed the experiments, authored or reviewed drafts of the article, and approved the final draft.
- Yu Qin analyzed the data, authored or reviewed drafts of the article, and approved the final draft.
- Yi Sun analyzed the data, authored or reviewed drafts of the article, and approved the final draft.
- Huifang Zhang analyzed the data, authored or reviewed drafts of the article, and approved the final draft.
- Jing Hu performed the experiments, analyzed the data, authored or reviewed drafts of the article, and approved the final draft.
- Kaiming Li performed the experiments, prepared figures and/or tables, and approved the final draft.
- Xuemei Yang performed the experiments, prepared figures and/or tables, and approved the final draft.

### Data Availability

The raw measurements are available in the Supplemental Files.

## PeerJ

## Supplemental Information

Supplemental information for this article can be found online at http://dx.doi.org/10.7717/peerj.17627#supplemental-information.

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
