# Peer review of "Exploring main soil drivers of vegetation succession in abandoned croplands of Minqin Oasis, China"

_PeerJ, doi:10.7717/peerj.17627_

## Round 0.1 · original submission · Major Revisions

The reviewers highlight that aspects of the experimental design and the statistical analysis are not clearly explained. Please respond carefully to their comments.

**Language Note:** The review process has identified that the English language must be improved. PeerJ can provide language editing services - please contact us at [email protected] for pricing (be sure to provide your manuscript number and title). Alternatively, you should make your own arrangements to improve the language quality and provide details in your response letter. – PeerJ Staff

Reviewer 1 ·

Basic reporting

1. This manuscript attempts to make some discussions on soil properties and vegetation succession in abandoned croplands of Minqin Oasis of China, but the treatments of experimental design is not suitable, and the paper is not innovative enough.
2. What are the causes of land abandonment? Establishment of protected areas? Scarcity of water? Or something else? The authors should make this clear, otherwise the reader will not understand the context of land use in the region.
3. The (Link between vegetation succession and soil properties) section does not present the results of the study in a concise manner, and is mostly a tedious presentation of the data analysis process.
4. Inouye et al. (1987), this literature is incorrectly cited. On the one hand, it is too old and there is a large amount of more relevant literature that can be cited. On the other hand, this MS studies abandoned land, whereas the subject of this literature is farmland.

Experimental design

Since the Minqin Oasis is divided into three typical areas, why was the study done only in the Hu area and not the wider area?

Validity of the findings

1. This manuscript attempts to make some discussions on soil properties and vegetation succession in abandoned croplands of Minqin Oasis of China, but the treatments of experimental design is not suitable, and the paper is not innovative enough.
2. The representativeness or significance of the study in this paper is not presented or discussed clearly, e.g., moisture, temperature, light, and even fire affect the successional process of vegetation at different scales.

Additional comments

Overall the references are old and it is recommended that authors try to add relevant literature from the last 5 years.

Reviewer 2 ·

Basic reporting

The overall language used in the manuscript is appropriate. However, there are a few sections where the language is hard to comprehend and where essential details are missing. For instance, the manuscript lacks sufficient information about the reasons behind using plant associations, the importance of doing so, the types of abandoned croplands, and the legacy story in detail. Additionally, the manuscript should maintain consistency in the use of terminology and formatting. For example, the plots are named differently in Figure 1 (numbers) and Figure 2 (letters), and in Table 1. Species names are sometimes in italics and sometimes in regular font. It is recommended that the captions for tables and figures should be concise, yet informative. It would be helpful to provide a complete description of the table and figure captions. The authors may want to consider merging Figure 1 and Figure 2 or moving Figure 2 to supplementary material. Only one picture is necessary for Figure 3. For Table 2, it is sufficient to use the term "Species" without including "Latin name". Although the raw data is available, there is no metadata, and some words are in Chinese. It would be beneficial if the authors could translate these words to English to make data more accessible.

Experimental design

The material and methods section in the current manuscript needs significant improvement to make the analysis replicable. Although the field sampling and soil sampling designs are somewhat supported by the figures, the authors need to provide clear explanations of their metrics and analysis, along with appropriate references and methodologies. For instance, it is unclear how the age of the abandoned croplands was determined. The authors must also provide a transparent workflow to determine all the metrics related to plant species and justify them. Additionally, the methods and units of the soil properties are missing, which need to be added. The data analysis section is also confusing; it is unclear whether the GAMs were performed as preliminary analysis or what the authors wanted to test. The authors should also explain why they used CCA to record the presence-absence.

Validity of the findings

We could expect the results regarding TS, this is not a novelty, but the authors have nicely discussed their findings and compared them with other studies, but what about the other soil properties found as drivers of the secondary succession?
Please, could the authors provide the results, performance and validation tests of the statistical analysis?
Please, authors should avoid long statements about factors that haven’t been included in the study (e.g. L305-312). Authors should highlight this as a limitation of their study.
The point 1 of your conclusion is indeed a result. Please, remove it from here.

---

## Round 0.2 · Minor Revisions

Reviewer 2 recognises that you have done a very thorough revision, and that your paper can have value for restoration. So please address their last comments, which will improve the readability and value of the paper.

Reviewer 2 ·

Basic reporting

Firstly, I acknowledge the authors for their impressive job improving the original manuscript. However, there is still some confusion about the hypothesis, the aims of the study and the final results that support (or not) the hypothesis. Evidence of this appears just at the beginning of the manuscript, i.e., the mismatch between the title, the abstract, and the last paragraph of the introduction. Please, amend it because this is critical to not lose the readers and most importantly, to be consistent with the scientific method.

In general, the references included have also been improved, though I have some suggestions for the authors regarding the previous research in the area:

L100 Could the authors include the references? “However, few studies have explored these connections.”

Experimental design

Please, read Section 1 about the research question. Being aware of limitations in resources and in representative areas with enough age range etc., it’s true that the study lacks power not including other areas different from the Hu area. I totally understand why the authors restricted their study to the Hu area but they should include the explanation of this limitation in the manuscript, and highlight potential research needs in their discussion.

A minor comment:
There is no need to include that you recorded the scientific names of the species, L146: where each shrub or herbaceous species was recorded using binomial nomenclature

Validity of the findings

If the aims of the study are focused only on salinity, then I must agree with #R1 that this study is not innovative enough.

I would suggest that authors do not mix between their conclusions and the implications of their study for management that indeed are quite well pointed out, e.g. 422 “These measures include the application of new irrigation methods, fertilization tactics, and plastic mulching.” Without any previous mention of this in the manuscript, this statement is out of the blue here, but it could be moved to the discussion section if the statement is supported with some references.

Additional comments

I would leave to the Editors the final decision about the innovativeness of this study to be published in the journal. However, I would like to highlight that, in land restoration, all data and interpretation of results, such as successional states, have scientific value.

---

## Round 0.3 · accepted · Accept

Thank you for making the requested changes.